# Investigating Molecular Mechanisms of Immunotoxicity and the Utility of ToxCast for Immunotoxicity Screening of Chemicals Added to Food

**DOI:** 10.3390/ijerph18073332

**Published:** 2021-03-24

**Authors:** Olga V. Naidenko, David Q. Andrews, Alexis M. Temkin, Tasha Stoiber, Uloma Igara Uche, Sydney Evans, Sean Perrone-Gray

**Affiliations:** Environmental Working Group, 1436 U Street NW, Suite 100, Washington, DC 20009, USA; dandrews@ewg.org (D.Q.A.); alexis@ewg.org (A.M.T.); tstoiber@ewg.org (T.S.); uloma.uche@ewg.org (U.I.U.); sydney.evans@ewg.org (S.E.); sean@ewg.org (S.P.-G.)

**Keywords:** immunotoxicology, multi-omics approaches in immunotoxicology, immunotoxic aspects of food additives, high-throughput screening, ToxCast, food additive, food contact substance, tert-butylhydroquinone, per- and polyfluoroalkyl substances

## Abstract

The development of high-throughput screening methodologies may decrease the need for laboratory animals for toxicity testing. Here, we investigate the potential of assessing immunotoxicity with high-throughput screening data from the U.S. Environmental Protection Agency ToxCast program. As case studies, we analyzed the most common chemicals added to food as well as per- and polyfluoroalkyl substances (PFAS) shown to migrate to food from packaging materials or processing equipment. The antioxidant preservative tert-butylhydroquinone (TBHQ) showed activity both in ToxCast assays and in classical immunological assays, suggesting that it may affect the immune response in people. From the PFAS group, we identified eight substances that can migrate from food contact materials and have ToxCast data. In epidemiological and toxicological studies, PFAS suppress the immune system and decrease the response to vaccination. However, most PFAS show weak or no activity in immune-related ToxCast assays. This lack of concordance between toxicological and high-throughput data for common PFAS indicates the current limitations of in vitro screening for analyzing immunotoxicity. High-throughput in vitro assays show promise for providing mechanistic data relevant for immune risk assessment. In contrast, the lack of immune-specific activity in the existing high-throughput assays cannot validate the safety of a chemical for the immune system.

## 1. Introduction

The immune system is one of the targets of chemical toxicity. Immune toxicity has been reported for a variety of substances, including arsenic compounds [1], chlorinated solvents [2], pesticides [3], and per- and polyfluoroalkyl substances [4], and the developing immune system is more susceptible to the harmful impact of chemical exposures compared to the adult immune system [5,6,7,8]. Immunotoxicology as a field has existed since the 1970s [9], and expert groups have published recommendations for conducting immune toxicity assessments [10,11]. Immunotoxicity is defined as the maladaptive functioning of the immune system following exposure to a xenobiotic substance. This phenomenon includes the loss of function (immunosuppression); excessive, damaging immune reactions (immunoenhancement); and alterations in the immune response that may be permanent or reversible (immunomodulation). The effect of a substance on the immune system depends on multiple factors such as route and duration of exposure and the toxicity mechanism. Immunotoxic effects can manifest in different ways, including lower antibody levels following vaccination, autoimmune symptoms, or systemic inflammation [12,13]. 

The recognition of the immune system’s susceptibility to environmental contaminants led to the development of assays that can evaluate chemicals for their immunotoxic potential [14]. Immunotoxicity analyses can evaluate histopathology and weight of immune system organs, lymphocyte counts, serum immunoglobulin levels, cell-mediated immune response, antibody production, natural killer (NK) cell function, and other functions [10]. Observational immunotoxicity measurements, such as analyses of the number and relative frequency of specific immune cell types, cannot provide mechanistic data on how the immune system is affected by a chemical substance. Functional tests such as T-cell-dependent antibody response assays and cytotoxicity assays advance immunotoxicity assessment one step further. Finally, the host resistance assays that measure immune defense against infectious agents have been considered the gold standard of immunotoxicity testing [15]. Given the complexity of the immune system, no single assay may be sufficient to determine immunotoxicity, and a combination of analyses for different immunological endpoints is likely needed for predicting immunotoxicity [12]. Mechanistic data, where available, can confirm the findings from other lines of evidence, such as studies in laboratory animals [13].

Despite the availability of testing methods for immunotoxicity, such testing has not been a priority in chemical risk assessment. The Organisation for Economic Co-operation and Development and European Union chemical test guidelines include an assessment of immune system parameters (such as lymphoid organ weights, hematology, and histopathology evaluation) as a part of standard oral 28-day and 90-day toxicity studies in laboratory animals, but do not require a systematic analysis of immunotoxicity for manufactured chemicals or contaminants [16]. The U.S. Environmental Protection Agency published immunotoxicity testing guidelines for pesticides in 1998, but later stated that this testing requirement can be waived [17,18]. The U.S. EPA Design for the Environment Program criterion document on hazard evaluation of chemical alternatives lists immunotoxicity among the less commonly considered toxicity endpoints [19]. 

New approach methodologies, such as high-throughput screening and omics technologies (genomics, transcriptomics, proteomics, and metabolomics), hold the promise of generating new data helpful for chemical risk assessment, including immunotoxicity assessment [20,21]. The U.S. EPA ToxCast program integrates high-throughput screening assays relevant to diverse toxicological endpoints, organ systems, and disease processes [22,23,24,25]. Two recent studies have classified ToxCast assays relevant to induction of chronic inflammation, an immune-mediated process [26,27]. However, a systematic evaluation of immune-relevant ToxCast assays is still needed. Due to the diversity of molecular pathways and orchestration of multiple cell types and tissues involved in the immune response, translation from high throughput data or animal-free models to toxicological endpoints relevant to the immune system remains a challenge [28]. 

Here we investigate whether the data generated under the U.S. EPA ToxCast program can be used for immunotoxicity screening. We focus on substances present in food, which include direct food additives and substances that can migrate into packaged food from food contact materials.

## 2. Materials and Methods

### 2.1. Data-Mining Strategy for the Identification of Immune-Relevant High-Throughput Assays 

Our analysis incorporates data from the U.S. EPA ToxCast CompTox Dashboard (https://comptox.epa.gov/dashboard, accessed on 24 September 2020) and the Comparative Toxicogenomics Database (http://ctdbase.org, accessed on 24 September 2020). The Comparative Toxicogenomics Database, developed by researchers from MDI Biological Laboratory (Salisbury Cove, ME, USA) and North Carolina State University (Raleigh, NC, USA), is a manually curated database based on peer-reviewed literature listed in PubMed [29,30]. Interactions for the phenotype described as “immune system process” were retrieved from the Comparative Toxicogenomics Database website in September 2020. 

ToxCast datasets were directly downloaded from the U.S. EPA CompTox Chemicals Dashboard, and all ToxCast information cited here represents the data viewable on the U.S. EPA website in September 2020. The ToxCast program comprises hundreds of high-throughput assays developed under different assay platforms, including both cell-free and cell-based assays [31]. Assay results for individual chemicals are classified by the ToxCast as hit-call “active” or “inactive” (meeting or not meeting dose–response criteria). Modeled AC_50_ values (activity concentration at 50% maximal activity) are viewable in the CompTox Dashboard. Each chemical is also assigned a “cytotoxicity limit”, a calculated value that reflects the overall cytotoxicity of an individual chemical in ToxCast assays [31]. The cytotoxicity limit does not indicate the cytotoxicity of a substance in a specific assay, and data from assays with AC_50_ values above the calculated cytotoxicity limit may be biologically relevant [27]. 

ToxCast data modeling published on the U.S. EPA CompTox Dashboard includes an assignment of data quality flags to assays such as “less than 50% efficacy”, “only highest [concentration] above baseline”, “borderline active”, “noisy data”, and other flags. Based on the manual review of ToxCast assay charts for the chemicals included in this study, we focused on two types of assays with the strongest evidence for response specificity: assays with no data quality flags and assays with only one data quality flag, “less than 50% efficacy”. 

Among the ToxCast assays activated by the case study compounds, we identified the assays that focused on a specific gene target (thus excluding assays that measure cytotoxicity, proliferation, or viability) and reviewed the function of each gene according to the information listed in the NCBI Gene database (https://www.ncbi.nlm.nih.gov/gene/, accessed on 24 September 2020) and PubMed database. We classified ToxCast assays as relevant to the immune system based on the involvement of their target genes or proteins in innate and adaptive immune responses, as reported in the peer-reviewed literature and gene function summaries. 

### 2.2. Identification of Case Study Compounds

#### 2.2.1. Direct Food Additives

Karmaus et al. [32,33] examined the universe of chemicals allowed in food in the United States for which data are available in the U.S. EPA ToxCast and identified a set of 556 chemicals directly added to food for functional purposes and 339 chemicals that may migrate to food from packaging, processing, or cleaning chemicals. Some chemicals in the above subset may be approved for use both as direct additives and in food packaging. We started with the list of 556 direct food additives defined by Karmaus et al. [33] along with the addition of butylated hydroxyanisole, a direct food additive used as an antioxidant preservative [34]. To identify the additives used most commonly in the U.S. food supply, chemical names for the 557 direct additives from Karmaus et al. [33] were matched to labels for over 120,000 packaged food and beverage products sold in U.S. grocery stores between 2018 and 2020. The label data analyzed by our research group was provided by Label Insight, a leading source of validated metadata for product attributes. Label Insight’s label data coverage exceeds 80% of consumer product goods sold in the United States (https://www.labelinsight.com/about, accessed on 24 September 2020).

#### 2.2.2. Indirect Additives: Per- and Polyfluoroalkyl Substances

For our analysis of indirect food additives, we narrowed our scope to the class of per- and polyfluoroalkyl substances (PFAS), chemicals used in food packaging and with known immunotoxicity for some members of the class [35]. To identify which PFAS were reported to migrate to food, we searched PubMed for relevant publications using the search query (“PFAS” OR “fluorotelomer” OR “fluorochemical” OR “perfluoroalkyl” OR “perfluorinated” OR “polyfluorinated” or “PFOA” or “perfluorochemicals” OR “fluorinated” OR “FTOH”) AND (“migration” OR “release” or “extractable”) AND “food”. In addition, the references within relevant publications were also reviewed for inclusion in our analysis. PFAS-based food contact substances were also reviewed in the U.S. Food and Drug Administration (U.S. FDA) database of Packaging & Food Contact Substances [36]. 

## 3. Results

### 3.1. Identification of Case Study Compounds

To identify the most common food additives for which high throughput ToxCast data are available, we matched the list of 557 direct additives [33] to ingredient labels of products sold in the United States in 2018–2020. From this group, 81 substances were identified on ingredient labels, and we ranked the additives by frequency of use. Among the remaining additives that could not be matched to ingredient labels, the majority (430 additives) are flavoring substances, which typically appear on the label under the name of artificial or natural flavors. Due to lack of disclosure for flavoring ingredients, we were unable to assess either their frequency of use or estimate dietary exposure. Eighteen additives that were present in less than 10 product labels were not included in further review (benzyl benzoate, betaine, butylene glycol, butylparaben, butyric acid, citral, dioctyl sodium sulphosuccinate, menthol, myristic acid, oleic acid, phenoxyethanol, piperine, resveratrol, sodium iodide, tannic acid, terpineol, theobromine, xylose). The 63 substances present on labels of more than 10 products were categorized by their functional class (Figure 1) and included in subsequent analyses.

In contrast to direct food additives, prioritization of indirect food additives by frequency of exposure is challenging, since these substances are not disclosed on the ingredient label. The U.S. FDA Inventory of Effective Food Contact Substance Notifications lists 1493 compounds that have been approved for use in the United States as food contact substances (with 71 of those listed as replaced) as of September 2020 [36]. Based on the U.S. FDA food contact substances inventory data from the past 10 years, on average, around 65 new food contact materials receive U.S. FDA approval every year. For the case study on indirect food additives, we focused on the family of per- and polyfluoroalkyl substances (PFAS) that have been used in food contact materials for decades [37,38] and are associated with toxicity to the immune system [35]. The PFAS-based materials have been used in sealing gaskets in food processing equipment, repeat-use plastics, non-stick coatings on cookware, and oil- and water-resistant coatings on paper and cardboard food packaging [35]. These PFAS materials, typically polymers, can contain and release monomeric PFAS as well as PFAS fragmentation products [39]. A recent study based on the data from the U.S. National Health and Nutrition Examination Survey reported an association between PFAS concentrations in the body and consumption of meals whose packaging may contain PFAS, such as fast foods, pizzas, and popcorn [40]. Some examples of PFAS structures that have been used in food contact materials are depicted in Figure 2. 

To identify PFAS species reported to end up in food from food contact materials, we compiled information from the studies of PFAS migration based on publications in PubMed (findings summarized in Appendix A
Table A1). Across all studies identified, PFAS migration varied depending on the type of food and its composition, with fatty materials typically enabling the greatest migration of PFAS from food packaging to food materials or food simulators, and short-chain PFAS migrating more readily compared to long-chain PFAS, a finding consistent with other reports in the literature [42]. Further, given that the laboratory methods are generally limited to detecting PFAS for which analytical standards are available, it is likely that some portion of PFAS that may migrate to food, by weight or types of compounds, was missed. In 2020, the U.S. FDA announced the voluntary phase-out of the group of food contact substances based on 6:2 fluorotelomers (Figure 2B), due to uncertainty about public health risks of these compounds [41].

### 3.2. Identification of ToxCast Active Assays

Of the 22 unique PFAS identified in migration studies (Appendix A
Table A1), nine had ToxCast data for the specific compounds or their salts (Table 1). These PFAS and the 63 direct additives were selected for further review (Table 2). Assay coverage in ToxCast varied from 238 assays for lithium perfluorooctanesulfonate and 250 assays for butylated hydroxyanisole to over 1000 assays for PFOA and PFOS. For our initial analysis, we included the ToxCast assays that were classified in the U.S. EPA CompTox Dashboard as “active” and that had modeled AC_50_ values (half-maximal activity) below the calculated cytotoxicity limit for the individual substance.

Among PFAS listed in Table 1, perfluoroundecanoic acid (PFUnDA), perfluorooctanesulfonic acid (PFOS), and perfluorooctanoic acid (PFOA) had the highest number of active ToxCast assays. Among direct food additives, three structurally related preservatives, tert-butylhydroquinone (TBHQ), propyl gallate, and propyl paraben, as well as food colorant FD&C Red 3 (also called erythrosine) had the greatest number of active assays with good data quality and AC_50_ below their cytotoxicity limits, suggesting their broad biological activity in the high-throughput assays (Table 2). The substances with the highest number of active assays were reviewed for potential immune system impacts. 

### 3.3. Identification of Immune-Specific Interactions in the Comparative Toxicogenomics Database

To query whether immune-relevant effects for these food additives were reported by prior studies, we reviewed the chemical–phenotype interactions for each substance identified in Table 1 and Table 2 by searching the Comparative Toxicogenomics Database [29] under “immune system process” and substance name. We also queried PubMed with a search query of “Substance name AND T cell OR B cell OR Natural Killer OR immune OR immunotoxicity”. References for the immune system interactions listed in the Comparative Toxicogenomics Database were manually reviewed and cross-checked against the studies identified from PubMed to confirm that the studies listed in the Comparative Toxicogenomics Database examined the direct effects of a substance on the immune-related parameters and functions. 

Vitamins showed multiple phenotype interactions related to the “immune system process” in the Comparative Toxicogenomics Database as well as numerous related publications in PubMed. This result was expected, since vitamins are essential for the immune system and other biological processes. From the rest of the direct additive group, three substances had “immune system process” interactions in the Comparative Toxicogenomics Database: sodium lauryl sulfate (two interactions), TBHQ (12 interactions), and silicon dioxide (89 interactions from studies of immune-mediated inflammatory responses following inhalation exposure to silica and related materials). Among the PFAS, PFOA had 35 recorded interactions for “immune system process”, and PFOS had three. 

A limited number or the lack of interactions in the Comparative Toxicogenomics Database does not mean that the substance does not affect a particular activity because testing might not have been done to assess that endpoint, or because the relevant studies, such as testing conducted by government agencies or chemical manufacturers, may not be included in the database. For example, Rice et al. reported the adverse effects of PFHxA and 6:2 FTOH on the immune system [43]. However, these compounds do not have any immune interactions in the Comparative Toxicogenomics Database. On the other hand, prior studies reported that TBHQ changes multiple immune parameters [44,45,46] and that PFOA suppresses the immune system and decreases antibody response to vaccines [47], and the Comparative Toxicogenomics Database documents immune-related interactions for both substances.

### 3.4. Analysis of Immune-Related Assays within ToxCast

We reviewed the active ToxCast assays for all compounds included in this study and observed that food colorant FD&C Red 3, antioxidant preservative TBHQ, and perfluoroundecanoic acid all showed activity towards multiple immune-related gene targets (Table 3). The targets included secreted proteins involved in cellular communication within the immune system (chemokines, cytokines, and growth factors); immune cell surface receptors (transmembrane proteins CD38, CD40, CD69, as well as leukotriene and prostaglandin receptors); and cell adhesion molecules that mediate the interaction between leukocytes and other cell types (E selectin, P selectin, ICAM1, and VCAM1). These proteins mediate the immune cell cross-talk and the orchestration of the innate and adaptive response against pathogens. 

Among the assays listed in Table 3, all but one come from the BioSeek assay platform in Toxcast. This high-throughput screening platform is based on primary human cells, and individual BioSeek assays may include endothelial cells, peripheral blood mononuclear cells, bronchial epithelial cells, or fibroblasts [48]. The BioSeek assays use antibodies to detect either increase (up) or decrease (down) in target protein expression [48]. The direction of the BioSeek assays affected by TBHQ, FD&C Red 3, and PFUnDA was “down”, indicating a decrease in the cell surface expression of target proteins. Since the BioSeek assay platform uses human cells, substance activity in these assays suggests potential relevance for the human immune system.

Perfluorodecanoic acid (PFDA) was active in some of the same immune-target assays as PFUnDA (Table 3). Surprisingly, PFOA, PFOS, and perfluorononanoic acid (PFNA) did not show strong activity. PFNA affected one assay targeting cytokine CXCL10. PFOS affected two targets, CXCL10 and HLA-DRα, major histocompatibility complex class II antigen-presenting molecule. PFOA, a well-studied PFAS with confirmed suppressive effects on the immune system, was weakly active in one immune-related assay, an assay targeting leukotriene B4 receptor, a transmembrane receptor involved in immune response and inflammation. Previous studies reported PFOA activity in ToxCast assays targeting estrogen receptor and peroxisome proliferator-activated receptors [26,49], indicating that ToxCast can identify active assays for this substance. No active assays for immune-related targets were identified for other PFAS listed in Table 1 (perfluoroheptanoic acid, perfluorohexanoic acid, 6:2 fluorotelomer alcohol, and 8:2 fluorotelomer alcohol). 

Finally, we identified the activity of TBHQ, FD&C Red 3, PFDA, and PFUnDA in a subset of assays targeting proteins involved in extracellular matrix remodeling, coagulation, and fibrinolysis, processes that are involved in innate and adaptive immune response, inflammation and host defense (Table 4). For this group of targets, FD&C Red 3 showed activity in most assays, followed by PFUnDA, TBHQ, and PFDA. PFOS affected the expression of one target, matrix metalloprotease 9. No other PFAS in our study affected the ToxCast targets listed in Table 4. 

The observed activity of TBHQ and PFUnDA in immune-specific ToxCast assays is consistent with the research literature. TBHQ has been reported to affect different immune system parameters and functions [44,45,46]. The immunosuppressive effects of PFUnDA have been reported in epidemiological studies [50,51], consistent with data for other PFAS. In contrast, the activity of FD&C Red 3 in assays with immune-specific targets was unexpected. No immune system process interactions were identified for this synthetic food colorant in the Comparative Toxicogenomics Database. Similarly, we could not identify studies in PubMed that reported immune system effects of FD&C Red 3. Given that this compound is active in a subset of immune-targeted assays in ToxCast, it would be important to test FD&C Red 3 in a standard panel of immunotoxicity assays.

TBHQ was also active in the ToxCast assays measuring the activity of transcription factors Nrf2 (nuclear factor, erythroid 2-like 2), aryl hydrocarbon receptor, and glucocorticoid receptor (Table 5). Prior studies reported that TBHQ effects are mediated via Nrf2, a transcription factor that regulates genes involved in antioxidant response, injury, and inflammation [52,53], and also via aryl hydrocarbon receptor (AhR) [54,55]. Aryl hydrocarbon receptor mediates cellular responses to a variety of xenobiotic substances and is now known to play an important regulatory role in the immune system [56]. The TBHQ-related activation of Nrf2 and AhR in ToxCast assays supports the specificity of TBHQ activity. PFOA, PFOS, and 6:2 FTOH also activated Nrf2 in ToxCast assays (Table 5).

Overall, we were surprised that PFAS with chain length shorter than 10 fluorinated carbons showed limited or no effect in immune-related assays in ToxCast because peer-reviewed scientific literature and authoritative agency assessments reported that multiple members of the PFAS class, including PFNA, PFOS, and PFOA as well as PFAS mixtures, show toxicity to the immune system both in laboratory animal studies and in human epidemiological studies [35,47].

### 3.5. Correlation Analysis of ToxCast and Immunological Data for TBHQ

In our analysis, TBHQ emerged as an example of a chemical with strong data availability both in high-throughput assays and in immunological studies. Based on PubMed searches, we identified specific molecular targets reported for TBHQ in studies published over the past decade (Appendix A
Table A2) and compared them to ToxCast data.

Immunological studies documented TBHQ’s influence on immune functions and parameters including altered T-cell, B-cell, and NK cell function. TBHQ’s effects reported in multiple studies include increased Nrf2 expression and transcriptional activity, a decrease in NFκB transcriptional activity, lower expression of the cell surface receptors CD69 and CD25, and inhibition of IL-2 and IFN-γ secretion [45,46,53,57,58], as summarized in Table 6. Additionally, Koh et al. [59] reported that TBHQ attenuated the increase in the production of pro-inflammatory mediators TNFα, IL-1β, IL-6, and prostaglandin E2. While most of the molecular targets of TBHQ reported in prior studies did not have corresponding ToxCast assays, six targets had such assays (Table 6).

Modeled AC_50_ concentrations for TBHQ activity in immune-specific assays in ToxCast, as reported in the CompTox Dashboard, were in the micromolar range, varying between 1.38 and 10.90 μM. Immunological studies of TBHQ reported activity in a similar dose range of 0.1–300 μM (Appendix A
Table A2). In our view, the concordance between TBHQ targets observed in ToxCast and in immunological studies (Table 6) increases the confidence in the potential of ToxCast to identify at least some immunotoxicity effects that would be relevant to human risk assessment. Studies are needed to analyze how the effects reported in high-throughput screening assays may translate to changes in specific immune functions such as defense against pathogens, autoimmunity, and anti-tumor immunity. 

## 4. Discussion

In 2020, the COVID-19 pandemic stimulated public and scientific attention to the environmental factors that can impact the immune system [60]. As a step towards this important goal, our study investigated the use of high-throughput ToxCast data for assessing immunotoxicity. We focused on chemicals added to food, a group well represented in ToxCast [32,33,61]. In addition to direct food additives, our study also included PFAS, substances that are not considered food additives, yet end up in food due to migration out of food contact materials [37,38,41]. Most food additives in the U.S. were approved decades ago and stayed on the market without a requirement for the manufacturers to conduct new toxicity studies [62]. While the U.S. FDA guidance for the safety review of food ingredients mentions immunotoxicity assessment [63], such testing is not required for previously approved additives. For food contact substances, U.S. FDA guidance mentions immunotoxicity studies only for substances with high daily exposure [64]. Recent publications noted that the total extent of exposure to food contact substances and their health and environmental effects remain unknown [65] and called for additional research on developmental immunotoxicity assessment of food contact materials [66]. 

Analyzing the ToxCast datasets for assay targets relevant to the immune system, we identified several distinct situations: (1) where in vitro screening does not show results expected from in vivo studies; (2) where in vitro screening data agree with immunological studies; and (3) where in vitro screening data point to a risk to the immune system that has not been previously reported and should be further investigated (Table 7).

For TBHQ and perfluoroundecanoic acid, ToxCast data indicate activation of multiple immune-related end points, in agreement with the available evidence from laboratory animal or epidemiological studies. In contrast, food colorant FD&C Red 3 has ToxCast data suggestive of immunotoxicity but lacks animal and epidemiological evidence, while PFOA does not show strong activity in ToxCast assays with immune targets yet has strong evidence of immunotoxicity in toxicological and epidemiological studies. Examples where high-throughput data are not in alignment with “classical” toxicological, epidemiological or immunological studies suggest that the current ToxCast datasets are not sufficient to confirm the lack of immunotoxicity. 

The data for FD&C Red 3 were particularly intriguing. Recently, Chappell et al. reported that this compound shows activity in several ToxCast assays relevant to neurodevelopmental processes [67]. In PubMed searches, we could not identify a study testing the immune system effects of this colorant. Strong activity of FD&C Red 3 on immune-related targets in ToxCast suggests to us that this endpoint should be analyzed further.

Based on the data showing TBHQ immune activity in animal and mechanistic studies, it is important to assess how TBHQ may affect the immune system in people. Despite the differences in murine and human immune systems, including immune cell population ratios and the presence or absence of certain chemokines, the majority of immune cell transcriptomes are conserved between mice and humans [68,69]. For TBHQ, we noted a concordance between immune targets identified in assays on mice or in murine cells (Appendix A
Table A2) and ToxCast targets in primary human cells used for the BioSeek assays (Table 6). More studies are needed to elucidate potential effects of TBHQ on immune parameters such as defense against infection, anti-tumor immune responses, and autoimmune reactivity. One recent study reported that intraperitoneal injection of TBHQ recruited innate immune cells and rendered mice less susceptible to mouse cytomegalovirus infection [53]. In contrast, another study reported that dietary exposure to TBHQ impaired NK cell cytotoxicity against influenza infection [44]. These distinct results may be related to the route of TBHQ exposure, the route of viral infection, or the skewing effects of TBHQ on the differentiation of T_h_1 and T_h_2 helper T cells [70]. The change in the T_h_1-T_h_2 cell balance represents an immunomodulatory effect that can affect distinct immunologic endpoints such as response against different types of pathogens, allergic response, and autoimmune conditions, all of which are complex, orchestrated outcomes that cross the definitions of immunosuppression and immunoenhancement.

Given the immunological activity of TBHQ, it is puzzling why this endpoint was previously overlooked. TBHQ was reviewed by the Joint FAO-WHO Expert Committee Report on Food Additives [71], the U.S. National Toxicology Program (NTP) [72], and the European Food Safety Authority [73], and none of these authoritative agencies highlighted the potential for TBHQ to harm the immune system. Reviewing the original studies cited in published assessments, we noted that TBHQ’s effects on the immune system were cited in those documents but did not receive further review or investigation. The NTP observed changes in the spleen of laboratory rodents exposed to TBHQ in the diet, such as the elevated incidence of splenic pigmentation known as hemosiderin. The NTP report described the effects on the spleen in the following way: “The pathogenesis of this change remains uncertain, and the biological significance was considered minimal” [72]. 

We also identified a two-week in vivo immunotoxicity study conducted in 1987 under contract with the NTP; while that study was not published in peer-reviewed literature, it was cited in the European Food Safety Authority report on TBHQ [73]. That study reported that TBHQ exposure increased spleen weight, decreased neutrophil counts, increased NK cell activity, increased serum complement C3 and increased Fc-mediated adherence and phagocytosis by peritoneal adherent cells. The study described these immune system changes as a “physiologic response” to TBHQ (cited in [73]). Such statements from earlier assessments illustrate how the risk of chemical toxicity to the immune system was disregarded.

The toxicity of PFAS to the immune system was demonstrated in epidemiological studies and toxicological experiments in laboratory animals [47]. The presence of PFAS in people, extensively documented through biomonitoring studies, correlates with attenuated antibody response to vaccinations in children and adults [35]. Some studies reported a correlation between PFAS levels in the body and lower resistance to disease or an increased risk of infections [74,75]. A relationship between higher PFAS levels and increased risk of asthma was reported [76] as well as an association between higher PFAS levels and increases in adolescent food allergies [77]. In 2020, the European Food Safety Authority identified the immunotoxicity of PFAS as the critical health effect [35]. 

In light of the epidemiological findings, the lack of immune-related effects in ToxCast for PFOA and other PFAS opens important questions for future research. The discrepancy between the behavior of certain PFAS in ToxCast and their effects in vivo may be related to the nature of PFAS interactions with cells and cellular processes. The exact mechanism of PFAS toxicity remains under research, although available data suggest the involvement of PPARα, NFκB and Nrf2 [78,79,80,81,82]. Previous studies reported PFAS effects on ToxCast assays targeting estrogen receptor, peroxisome proliferator-activated receptors (PPAR) alpha and gamma, pregnane X receptor, and androgen receptor [26]. The investigation of PPARα-independent mechanisms of PFAS toxicity also suggests suppression of STAT5B [49], a transcription factor activated by cytokines and involved in the immune cell development and autoimmunity [83]. Further, there is a growing body of research on the cross-talk between the endocrine and the immune systems [84], and some impacts of PFAS may be endocrine-mediated. Finally, ToxCast data summarized in Table 5 indicate that 6:2 FTOH, PFOA, and PFOS were active in assays targeting the transcription factor Nrf2. Recent studies have identified Nrf2 as a mediator of PFAS toxicity and signaling in mouse testis and liver as well as frog liver [79,80,81]. However, PFOA’s effect on Nrf2-mediated pathways seems to be distinct from other established Nrf2 activators [82].

Structurally, PFAS examined here are similar to fatty acids with carbon-fluorine bonds instead of carbon-hydrogen bonds. Due to their physicochemical nature, PFAS repel both water and lipids. PFAS bind to protein targets, such as serum albumin [85], peroxisome proliferator-activated receptor [86], and estrogen receptor [87]. PFAS also interact with, insert into, and disrupt phospholipid bilayer and model membranes [88,89]. Hypothetically, the membrane-disrupting effects of PFAS may translate into an impact on the physiological processes that depend on transmembrane receptors such as immune recognition and immune defense against pathogens. Future mechanistic studies of PFAS interaction with lymphocytes and other cell types may be able to address this hypothesis and shed light on the molecular mechanisms of PFAS immunotoxicity.

The PFAS example shows the limitations of the currently available high-throughput assays for immunotoxicity screening. The existing assays likely do not capture the full extent of the possible mechanisms of immunotoxicity, especially since different subpopulations of immune cells play distinct roles in the immune defense against different infectious agents and anti-tumor immunity [90]. The lack of ToxCast activity does not indicate that a substance does not impact a particular biological system, such as the immune system, because assays for a specific outcome or a toxicological endpoint might not be in ToxCast as yet. In contrast, the TBHQ analysis supports the value of such approaches, since a strong correlation is seen between high-throughput screening data for TBHQ and findings from immunological assays. Future immunotoxicity studies of FD&C Red 3 may provide further information for probing this relationship between different types of data and verifying the ToxCast results for this substance.

## 5. Conclusions

Joint consideration of toxicological and high-throughput screening data suggests that chemicals directly or indirectly added to food for decades—such as PFAS and TBHQ—may show previously unanticipated effects on the immune system. From the public policy perspective, the discovery of impacts on human health of substances that have long been used in consumer products and food products suggests that the pre-market safety evaluation of these substances was inadequate. We recommend that immunotoxicity testing should be prioritized in order to protect public health, and immunotoxicity analysis should be, in our estimate, an integral part of chemical safety assessment.

## Figures and Tables

**Figure 1 ijerph-18-03332-f001:**
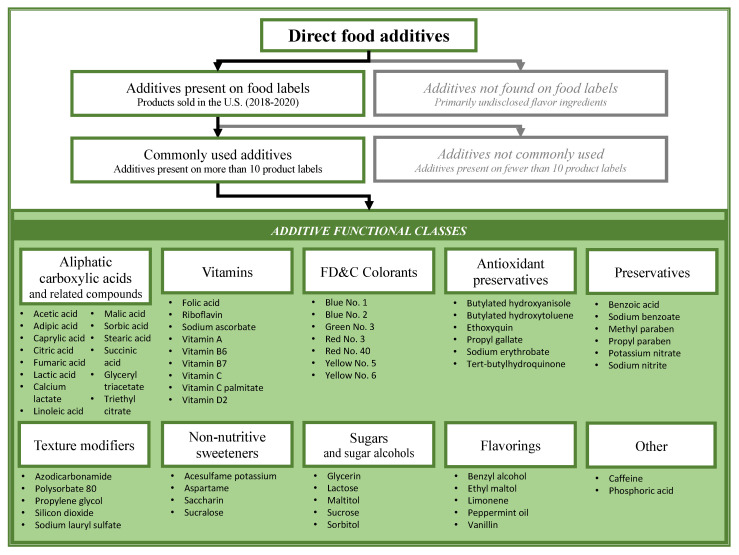
Flow diagram for identification of direct additives included in this study. Initial group of direct food additives analyzed in this study was defined in a publication by Karmaus et al. [33].

**Figure 2 ijerph-18-03332-f002:**
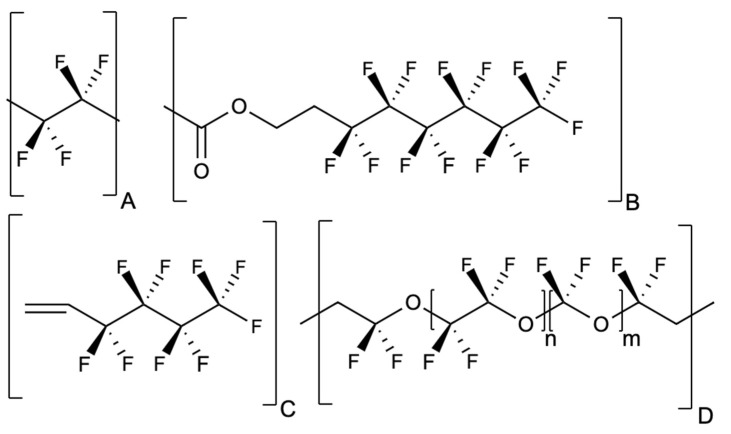
Structures of several PFAS food contact materials. (**A**) shows the structure of polytetrafluoroethylene or PTFE, used for coatings on cookware, pans, and utensils. (**B**) shows the 6:2 fluorotelomer structure present in multiple food contact substances approved by the U.S. FDA since 2008, which are undergoing the voluntary phase-out starting in July 2020 [41]. (**C**) shows the fluorinated monomer ingredient that, in combination with perfluoroethylene and ethylene, is used to manufacture a PFAS-based terpolymer approved in 2018 (U.S. FDA food contact notification approval No. 1914). (**D**) shows the fluorinated section of a perfluoroether polymer approved in 2010 (U.S. FDA food contact notification approval No. 962).

**Table 1 ijerph-18-03332-t001:** Number of ToxCast assays for per- and polyfluoroalkyl substances (PFAS) with the half-maximal activity concentration (AC_50_) below the cytotoxicity limit.

PFAS Reported to Migrate to Food, with CAS Numbers	Number of Assays with Half-Maximal Activity Concentration (AC_50_) < Cytotoxicity Limit
Perfluorooctanesulfonic acid (1763-23-1)	48
Perfluoroundecanoic acid (2058-94-8)	45
Perfluorooctanoic acid (335-67-1)	41
Perfluorohexanoic acid (307-24-4)	22
Perfluorodecanoic acid (335-76-2)	18
Potassium perfluorooctanesulfonate (2795-39-3)	13
Ammonium perfluorooctanoate (3825-26-1)	11
Perfluorononanoic acid (375-95-1)	9
Perfluoroheptanoic acid (375-85-9)	5
6:2 fluorotelomer alcohol (647-42-7)	4
8:2 fluorotelomer alcohol (678-39-7)	1
6:2 fluorotelomer methacrylate (2144-53-8)	1
Lithium perfluorooctanesulfonate (29457-72-5)	0

Note: Based on ToxCast data publicly available in September 2020. This table includes assays without any data quality flags and assays with a single data quality flag, “less than 50% efficacy”.

**Table 2 ijerph-18-03332-t002:** Number of ToxCast assays for direct food additives with the half-maximal activity concentration (AC_50_) below the cytotoxicity limit identified in Figure 1.

Direct Food Additives	Number of Assays with Half-Maximal Activity Concentration (AC_50_) < Cytotoxicity Limit
Tert-butylhydroquinone (TBHQ)	58
FD&C Red No. 3 (erythrosine)	46
Propyl paraben	23
Propyl gallate	21
Ethoxyquin, FD&C Blue No. 1, folic acid, sodium lauryl sulfate, sorbic acid, vitamin D2	11–16
Acetic acid, caprylic acid, FD&C Green No. 3, maltol, methyl paraben, sodium ascorbate, stearic acid, triethyl citrate, vitamin A	6–10
Acesulfame potassium, adipic acid, ascorbyl palmitate, aspartame, azodicarbonamide, benzoic acid, benzyl alcohol, butylated hydroxytoluene, caffeine, calcium lactate, citric acid, ethyl maltol, FD&C Red No. 40, FD&C Yellow No. 5, FD&C Yellow No. 6, fumaric acid, glycerin, glyceryl triacetate, limonene, linoleic acid, malic acid, peppermint oil, phosphoric acid, potassium nitrate, propylene glycol, riboflavin, saccharin, silicon dioxide, sodium erythorbate, sodium nitrite, sorbitol, sucralose, sugar, vanillin, vitamin B6, vitamin C	1–5
Butylated hydroxyanisole, FD&C Blue No. 2, lactic acid, lactose, polysorbate 80, sodium benzoate, succinic acid, vitamin B7	0

Note: Based on ToxCast data publicly available in September 2020. This table includes assays without any data quality flags and assays with a single data quality flag, “less than 50% efficacy”.

**Table 3 ijerph-18-03332-t003:** Immune targets in ToxCast assays affected by tert-butylhydroquinone, FD&C Red 3, and several per- and polyfluoroalkyl substances.

Gene Name and Function	ToxCast Assay Name	TBHQ	FD&C Red 3	PFOS	PFOA	PFNA	PFDA	PFUnDA
CCL2 (chemokine (C-C motif) ligand 2) Chemokine with chemotactic activity for monocytes and basophils	BSK_3C_MCP1_down		✓					
	BSK_CASM3C_MCP1_down		✓					
	BSK_KF3CT_MCP1_down		✓				✓	✓
	BSK_LPS_MCP1_down	✓	✓					
	BSK_SAg_MCP1_down	✓						
CCL26 (chemokine (C-C motif) ligand 26) Chemokine with chemotactic activity for eosinophils and basophils	BSK_4H_Eotaxin3_down	✓						
CD38 molecule Transmembrane receptor expressed on macrophages, dendritic cells, and NK cells	BSK_SAg_CD38_down	✓						
CD40 molecule Transmembrane receptor expressed on antigen-presenting cells	BSK_LPS_CD40_down	✓						
	BSK_SAg_CD40_down	✓						
CD69 molecule Transmembrane receptor expressed on activated T cells and NK cells	BSK_SAg_CD69_down	✓						
CSF1 (macrophage colony-stimulating factor) Cytokine that controls the differentiation and function of macrophages	BSK_hDFCGF_MCSF_down	✓	✓					✓
	BSK_LPS_MCSF_down	✓	✓					
CXCL10 (chemokine (C-X-C motif) ligand 10) Chemokine involved in the stimulation of monocytes, NK cells, and T cells	BSK_BE3C_IP10_down			✓		✓	✓	✓
	BSK_hDFCGF_IP10_down	✓						✓
	BSK_KF3CT_IP10_down			✓			✓	✓
CXCL8 (chemokine (C-X-C motif) ligand 8) Chemokine secreted by macrophages, neutrophils, eosinophils, and T cells	BSK_hDFCGF_IL8_down	✓						✓
	BSK_LPS_IL8_down	✓						
	BSK_SAg_IL8_down	✓						
CXCL9 (chemokine (C-X-C motif) ligand 9) Chemokine involved in immune regulation and inflammatory processes	BSK_BE3C_MIG_down						✓	✓
	BSK_hDFCGF_MIG_down	✓						✓
	BSK_SAg_MIG_down	✓						
HLA-DRA (major histocompatibility complex class II) Antigen-presenting molecule	BSK_3C_HLADR_down		✓					
	BSK_BE3C_HLADR_down		✓	✓			✓	✓
ICAM1 (intercellular adhesion molecule 1) Cell adhesion molecule	BSK_KF3CT_ICAM1_down							✓
IL-1α (Interleukin-1, alpha) Cytokine produced by macrophages	BSK_BE3C_IL1a_down		✓				✓	✓
	BSK_KF3CT_IL1a_down		✓					✓
	BSK_LPS_IL1a_down	✓						
LTB4R (leukotriene B4 receptor) Transmembrane receptor on immune cells	NVS_GPCR_gLTB4				✓		✓	✓
E-selectin Cell adhesion molecule	BSK_LPS_Eselectin_down	✓	✓ *					
	BSK_SAg_Eselectin_down	✓						
P-selectin Cell adhesion molecule	BSK_4H_Pselectin_down	✓						
Prostaglandin E receptor 2 Transmembrane receptor on immune cells	BSK_LPS_PGE2_down	✓	✓					
TGF-β1 (transforming growth factor, beta 1) Growth factor that regulates immune responses	BSK_BE3C_TGFb1_down		✓				✓	✓
	BSK_KF3CT_TGFb1_down	✓	✓					✓
TNF (tumor necrosis factor) Pro-inflammatory cytokine primarily secreted by macrophages	BSK_LPS_TNFa_down	✓						
VCAM1 (vascular cell adhesion molecule 1) Cell adhesion molecule	BSK_3C_VCAM1_down	✓						
	BSK_4H_VCAM1_down	✓						
	BSK_hDFCGF_VCAM1_down	✓						✓
	BSK_LPS_VCAM1_down	✓						

Note: * This assay had one data quality flag indicating “noisy data”.

**Table 4 ijerph-18-03332-t004:** ToxCast assays targeting extracellular matrix remodeling, coagulation, and fibrinolysis.

Gene Name and Function	ToxCast Assay Name	TBHQ	FD&C Red 3	PFOS	PFDA	PFUnDA
Coagulation factor III Involved in the innate immune response and host defense against infection, initiates the coagulation cascades	BSK_LPS_TissueFactor_down	✓				
Matrix metallopeptidase 1 Enzyme that breaks down extracellular matrix; involved in the immune response to infection	BSK_BE3C_MMP1_down		✓			
	BSK_hDFCGF_MMP1_down	✓	✓			✓
Matrix metallopeptidase 9 Enzyme that breaks down extracellular matrix; involved in the immune response to infection	BSK_KF3CT_MMP9_down		✓	✓	✓	✓
Tissue plasminogen activator Secreted serine protease that converts the plasminogen to plasmin, regulates innate immune response	BSK_BE3C_tPA_down		✓		✓	✓
Urokinase-type plasminogen activator Secreted serine protease that converts the plasminogen to plasmin, regulates innate immune response	BSK_BE3C_uPA_down		✓			✓
	BSK_KF3CT_uPA_down		✓			✓
Urokinase-type plasminogen activator receptor Regulates inflammatory, immune, and coagulation responses	BSK_3C_uPAR_down		✓			
	BSK_BE3C_uPAR_down		✓			
	BSK_CASM3C_uPAR_down		✓			
SERPINE1 (serpin peptidase inhibitor, clade E) Involved in the innate immune response and early host defense against infection, an inhibitor of fibrinolysis	BSK_BE3C_PAI1_down		✓		✓	✓
	BSK_hDFCGF_PAI1_down	✓	✓			✓
Thrombomodulin Transmembrane receptor, suppresses coagulation and inflammation	BSK_CASM3C_Thrombomodulin_down		✓			
	BSK_CASM3C_Thrombomodulin_up	✓				
TIMP metallopeptidase inhibitor 1 Inhibitor of the enzymes that break down extracellular matrix; regulates innate immune response	BSK_hDFCGF_TIMP1_down	✓	✓			✓
TIMP metallopeptidase inhibitor 2 Inhibitor of the enzymes that break down extracellular matrix; regulates innate immune response	BSK_KF3CT_TIMP2_down		✓			✓

**Table 5 ijerph-18-03332-t005:** Transcription factors affected by tert-butylhydroquinone and other compounds in ToxCast assays.

Gene Name and Function	ToxCast Assay Name	TBHQ	FD&C Red 3	6:2 FTOH	PFOS	PFOA	PFUnDA
Aryl hydrocarbon receptor (AhR) Involved in xenobiotic response	ATG_Ahr_CIS_up	✓					
	OX21_AhR_LUC_Agonist *	✓					
Nuclear factor, erythroid 2-like 2 (NFE2L2, Nrf2) Involved in oxidative stress, inflammation, and injury	ATG_NRF2_ARE_CIS_up	✓		✓	✓	✓	
	TOX21_ARE_BLA_agonist_ratio	✓				✓	
Glucocorticoid receptor (nuclear receptor subfamily 3, group C, member 1) Involved in regulation of stress response, inflammation, and immune processes	NVS_NR_hGR				✓		✓
	TOX21_GR_BLA_Antagonist_ratio	✓					

Note: * This assay had an AC_50_ value greater than the calculated cytotoxicity limit for TBHQ.

**Table 6 ijerph-18-03332-t006:** Cellular and molecular targets affected by TBHQ that have ToxCast assays.

TBHQ Target Reported in Immunological Studies	Studies Reporting This Target	ToxCast Assay Direction
Increased activity of Nrf2	[45,46,53]	up
Decreased activity of NFκB	[46,58]	down *
Decreased CD69 expression	[45,57]	down
CCL2 increase	[53]	down
TNFα decrease	[59]	down
IL-6 decrease	[59]	no activity

Note: * Active assay with several data quality flags identified in ToxCast.

**Table 7 ijerph-18-03332-t007:** Summary of findings comparing high-throughput data with other data types.

Chemical	ToxCast	Laboratory Animal Studies	Epidemiological Studies	Conclusion
FD&C Red 3	Affects multiple immune parameters	No studies identified	No studies identified	Potential for immunotoxic effects, should be further investigated
TBHQ	Affects multiple immune parameters	Immune modulation, changes in the immune functions	No studies identified	Immunological and mechanistic studies point to risk for the immune system
PFUnDA	Affects multiple immune parameters	Some evidence of immune suppression	Immune suppression	Human data and mechanistic studies point to risk for the immune system
PFOA	Does not show strong activity in ToxCast assays with immune targets	Immune suppression	Immune suppression	Human data point to risk for the immune system with limited support from mechanistic studies

## Data Availability

This analysis is based on open access datasets from the U.S. EPA ToxCast (https://comptox.epa.gov/dashboard, accessed on 24 September 2020), and the Comparative Toxicogenomics Database (http://ctdbase.org, accessed on 24 September 2020).

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
