# Peer review of "Investigating Molecular Mechanisms of Immunotoxicity and the Utility of ToxCast for Immunotoxicity Screening of Chemicals Added to Food"

_ijerph, 2021, doi:10.3390/ijerph18073332_

Round 1

Reviewer 1 Report

The Authors investigated whether the data generated under the U.S. EPA ToxCast program can be used for immunotoxicity screening focusing on food additives and substances shown to migrate into food from packaging materials or processing equipment.

As the Authors stated, a limited number or the lack of interactions in the Comparative Toxicogenomics Database does not mean that the substance does not affect a particular activity because testing might not have been done to assess that endpoint, or because the relevant studies, such as testing conducted by government agencies or chemical manufacturers, were not included.

The the paper is interesting and well written. The conclusions sound.

References should be rewritten since reference 40 and 58 are not reported in the text.

Line 198: spelling error

Line 330: punctuation should be corrected.

Author Response

Author response: Authors thank the reviewer for interest in this research project and for the encouraging comment.

“References should be rewritten since reference 40 and 58 are not reported in the text.”

Author response: References have been updated, and references 40 and 58 are listed in the revised manuscript. Reference 40 appears on line 190 and reference 58 on line 354 of the revised manuscript (in track change mode). Reference 58 also appears in Table 6 and Table B1.

Line 198: spelling error

Author response: Sentence was corrected.

Line 330: punctuation should be corrected.

Author response: Sentence was corrected.

Reviewer 2 Report

In this manuscript Naidenko et al reported that we need to be cautious to use high-throughput assays to assess the immunotoxicity of chemicals because there is apparent lack of immune-specific activity in the existing ToxCast assays.  Some questions should be addressed:

  1. What cell lines are been used in ToxCast assays? Mouse vs. rat vs human cells; Hepatocytes, primary cells from Spleen or Thymus? This information should be provided in the manuscript.
  2. The authors have not addressed the effects of concentrations that were used in ToxCast assays: toxic concentration or concentrations more closed to human exposure
  3. In general, immune response is systemic effects, requiring multiple tissues and cell types. The authors should address this.
  4. The effects of PFAS agents are primarily mediated by PPAR activation, but not Nrf2. The authors may discuss whether and how these signaling pathways are differently activated in the in vivo and in vitro conditions

Author Response

  1. What cell lines are been used in ToxCast assays? Mouse vs. rat vs human cells; Hepatocytes, primary cells from Spleen or Thymus? This information should be provided in the manuscript.

Author response: ToxCast in vitro assays use a variety of both cell-free (biochemical assays) and cell-based assay platforms and cell lines. Example of cell-based assays include human primary cells and cell lines (such as HepG2 hepatocyte carcinoma) and rat primary liver cells. Human primary cells are used for assays under the ToxCast BioSeek platform. While these primary cells are not from spleen or thymus, the BioSeek assays include peripheral blood mononuclear cells  which are relevant for immune toxicity assessment.

Newly added and/or revised text on this topic is noted below, together with the line numbers in the manuscript with track changes.

Lines 101-103, “The ToxCast program comprises hundreds of high-throughput assays developed under different assay platforms, including both cell-free and cell-based assays [31]”

Lines 286-289, “The majority of assays identified in Table 3 come from the BioSeek assay platform in Toxcast. This high-throughput platform is based on primary human cells, and individual BioSeek assays may include endothelial cells, peripheral blood mononuclear cells, bronchial epithelial cells, or fibroblasts [48].”

Lines 292-294, “Since the BioSeek assay platform is based on human cells, substance activity in these assays suggests relevance for the human immune system.”

Lines 408-410, “For TBHQ, we saw general concordance between immune targets identified in assays on mice or in murine cells (Table B1) and ToxCast targets in primary human cells (the BioSeek assays).”

  1. The authors have not addressed the effects of concentrations that were used in ToxCast assays: toxic concentration or concentrations more closed to human exposure.

Author response: The topic of human exposure to TBHQ is an important and complex question. The revised manuscript highlights the similarities between reported AC50 concentrations from ToxCast assays and concentrations reported in the peer-reviewed literature where similar biological effects were observed.

Lines 361-364, “Modeled AC50 concentrations for TBHQ activity in immune-specific assays in ToxCast, as reported in the CompTox Dashboard, were in the micromolar range, varying between 1.38-10.90 uM. Immunological studies of TBHQ reported activity in a similar dose range of 0.1-300 uM (Table B1). In our view, the concordance between TBHQ targets observed in ToxCast and in immunological studies (Table 6) increases the confidence in the potential of ToxCast to identify some immunotoxicity effects that would be relevant to human risk assessment.”

  1. In general, immune response is systemic effects, requiring multiple tissues and cell types. The authors should address this.

Author response: We thank the reviewer for emphasizing this important point and we addressed this topic in several places in the revised manuscript. Newly added and/or revised text on this topic is noted below, together with the line numbers (in the manuscript with track changes).

Lines 57-59: “Given the complexity of the immune system, no single assay may be sufficient to determine immunotoxicity, and a combination of analyses for different immunological endpoints is likely needed for predicting immunotoxicity [12].”

Lines 80-83: “ Due to the diversity of molecular pathways and orchestration of multiple cell types and tissues involved in the immune response, translation from high throughput data or animal-free models to toxicological endpoints relevant to the immune system remains a challenge [28].

Lines 416-422, “These distinct results may be related to the route of TBHQ exposure, the route of viral infection, or the skewing effects of TBHQ on the differentiation of Th1 and Th2 helper T cells [70]. The change in the Th1-Th2 cell balance represents an immunomodulatory effect that can have distinct consequences for varying immunologic endpoints such as response against different types of pathogens, allergic response and autoimmune conditions, all of which are complex, orchestrated outcomes that cross the definitions of immunosuppression and immunoenhancement.”

Lines 487-489: “The existing assays likely do not capture the full extent of the possible mechanisms of immunotoxicity, especially since different subpopulations of immune cells play distinct roles in the immune defense against different infectious agents and anti-tumor immunity [90].”

  1. The effects of PFAS agents are primarily mediated by PPAR activation, but not Nrf2. The authors may discuss whether and how these signaling pathways are differently activated in the in vivo and in vitro conditions.

Author response: Thank you for the comment, we have included a paragraph and several new references to discuss existing research literature on Nrf2 and PFAS.

Revised text on lines 459-475 “In light of the epidemiological findings, the lack of immune-related effects in ToxCast for PFOA and other PFAS opens important questions for future research. The discrepancy between the behavior of certain PFAS in ToxCast and their effects in vivo may be related to the nature of PFAS interactions with cells and cellular processes. The exact mechanism of PFAS toxicity remains under research, although available data suggest the involvement of PPARa, NFkB and Nrf2 [78-82]. Previous studies reported PFAS effects on ToxCast assays targeting estrogen receptor, peroxisome proliferator-activated receptors (PPAR) alpha and gamma, pregnane X receptor, and androgen receptor [26]. The investigation of PPARa-independent mechanisms of PFAS toxicity also suggests suppression of STAT5B [49], a transcription factor activated by cytokines and involved in the immune cell development and autoimmunity [83]. Further, there is a growing body of research on the cross-talk between the endocrine and the immune systems [84] and some impacts of PFAS may be endocrine-mediated. Finally, ToxCast data summarized in table 5 indicate that 6:2FTOH, PFOA and PFOS were active in assays targeting the transcription factor Nrf2. Recent studies have identified Nrf2 as a mediator of PFAS toxicity and signaling in mouse testis and liver as well as frog liver [79-81]. However, PFOA effect on Nrf2-mediated pathways seem to be distinct from other established Nrf2 activators [82].”

Reviewer 3 Report

Although “Investigating Molecular Mechanisms of Immunotoxicity and the Utility of ToxCast for Immunotoxicity Screening of Chemicals Added to Food" has been extensively studied before, this article is the most comprehensive compilation of references in this area. The authors have done very good job to cover up to the latest work. I have no doubt that this paper will be much cited. This is a useful and interesting article and I think it makes a potent scientific contribution to environmental protection safety. It is an interesting paper, well written and structured, worth to be published in IJERPH. This manuscript might be acceptable in present form. 

Author Response

Author response: Authors thank the reviewer for taking the time to read the manuscript and for the supportive comment. Authors also bring to reviewer’s attention that small editorial changes have been made in the revised manuscript for better readability and clearer presentation of our findings.

Reviewer 4 Report

This study investigated the immunotoxicity of chemicals added to food by the use of ToxCast program. The screening progress was comprehensive and sufficient, which included direct food additives and indirect additives. The results were also cross-checked with those from previously published articles. Furthermore, the mechanisms of potential candidates were discussed sufficiently in the end. I am surprised that FD&C Red 3 has a strong potential to exhibit immunotoxicity. I think this compound deserves more research in the future. It is also interesting to know that ToxCast was not able to filter out the immunotoxicity of PFAS. I agree with the authors that incorporating new research data is essential for ensuring the safety of food additives. The methodology of this study can be further applied to other fields (e.g. textile or infant formula). Anyway, this manuscript is well designed and written. I strongly recommend publishing it soon.

Author Response

(The authors gave the same response as above.)

Round 2

Reviewer 2 Report

The authors have successfully addressed my comments.  No more comment.